Corrected: Publisher correction

# Lithospheric foundering and underthrusting imaged beneath Tibet

Min Chen[1], Fenglin Niu[1,2], Jeroen Tromp[3,4], Adrian Lenardic[1], Cin-Ty A. Lee[1], Wenrong Cao[1] & Julia Ribeiro[1]

Long-standing debates exist over the timing and mechanism of uplift of the Tibetan Plateau and, more specifically, over the connection between lithospheric evolution and surface expressions of plateau uplift and volcanism. Here we show a T-shaped high wave speed structure in our new tomographic model beneath South-Central Tibet, interpreted as an upper-mantle remnant from earlier lithospheric foundering. Its spatial correlation with ultrapotassic and adakitic magmatism supports the hypothesis of convective removal of thickened Tibetan lithosphere causing major uplift of Southern Tibet during the Oligocene. Lithospheric foundering induces an asthenospheric drag force, which drives continued underthrusting of the Indian continental lithosphere and shortening and thickening of the Northern Tibetan lithosphere. Surface uplift of Northern Tibet is subject to more recent asthenospheric upwelling and thermal erosion of thickened lithosphere, which is spatially consistent with recent potassic volcanism and an imaged narrow low wave speed zone in the uppermost mantle.

[1] 318 Keith-Wiess Geology Lab, Department of Earth Science, Rice University, MS 126, 6100 Main Street, Houston, Texas 77005, USA. [2] State Key Laboratory of Petroleum Resource and Prospecting, and Unconventional Natural Gas Institute, China University of Petroleum, Beijing 102249, China. [3] Department of Geosciences, Princeton University, Princeton, New Jersey 08544, USA. [4] Program in Applied and Computational Mathematics, Princeton University, Princeton, New Jersey 08544, USA. Correspondence and requests for materials should be addressed to M.C. (email: Min.Chen@rice.edu).

Since the onset of 'hard' continent–continent collision at about 45 Ma[1], India-Eurasia convergence has produced the highly elevated Tibetan Plateau and Himalayan Mountain Belt. Geodetic observations suggest not only inter-plate convergence between India and Eurasia, but also intra-plate deformation within Tibet at present[2,3]. Measurements based on the Global Positioning System indicate that current crustal motion relative to stable Eurasia decreases northwards, with rates of ~40 mm per year at Northern India, ~25 mm per year at Central Tibet and ~10 mm per year at Northern Tibet[2] (Fig. 1), and an ongoing convergence rate of ~20 mm per year between India and the Indus-Yarlung Suture (IYS) (the southern boundary of Tibet)[3], suggesting intra-plate shortening has to accommodate the velocity differences. Existing magnetic, paleomagnetic and volumetric balancing studies estimate that India-Eurasia convergence varies along the Himalayan arc, increasing from 1,800 km in the west to 2,800 km in the east[4]. Lithospheric processes that accommodate the total convergence have been widely speculated upon. Hypotheses include the following, potentially co-existing, scenarios: wholesale underthusting of the Indian plate beneath the plateau[5], underthrusting in the south with lower-crustal flow beneath the central and northern plateau[6,7], distributed shortening and thickening of the Tibetan crust[8], northward injection of the Indian crust[9], convective removal of the lower portion of thickened Tibetan lithosphere (TL)[10], indentation by a rigid Indian plate and continental block extrusion[11,12] and

intracontinental subduction from the India and/or the Asia side[13,14].

Surface-observation-based geodetic, geologic and tectonic studies often focus on the convergence budget of crustal lithosphere[15]. In contrast to well-agreed upon crustal thickening, direct evidence for thickening of the underlying lithospheric mantle has, thus far, been lacking. As such, the question of how continental mantle lithosphere is consumed beneath Tibet and its role in Tibetan tectonic evolution remains open[13]. Seismic tomographic images can map the geometry and spatial extent of mantle lithosphere in terms of high wave speed anomalies and shed light on Tibetan tectonic evolution[16,17]. However, differences among existing seismic images complicate the interpretation of mantle lithospheric process. For example, such processes under Central and Eastern Tibet are still heavily debated, and include underplating[5,18], high-angle subduction[19–21], horizontal extrusion[17] and distributed thickening together with subsequent convective removal[22]. Resolutions to these outstanding debates rely on more robust seismic observations of mantle lithosphere beneath Tibet, constraining, for example, the northern extent and convergence angle of Indian lithosphere (IL).

Discrepancies among previous tomographic studies are due to the following main causes, namely differences in station coverage or seismic phase information used in the inversion and different underlying theories such as the ray, normal mode and finite-frequency theories. The latest method developed for seismic

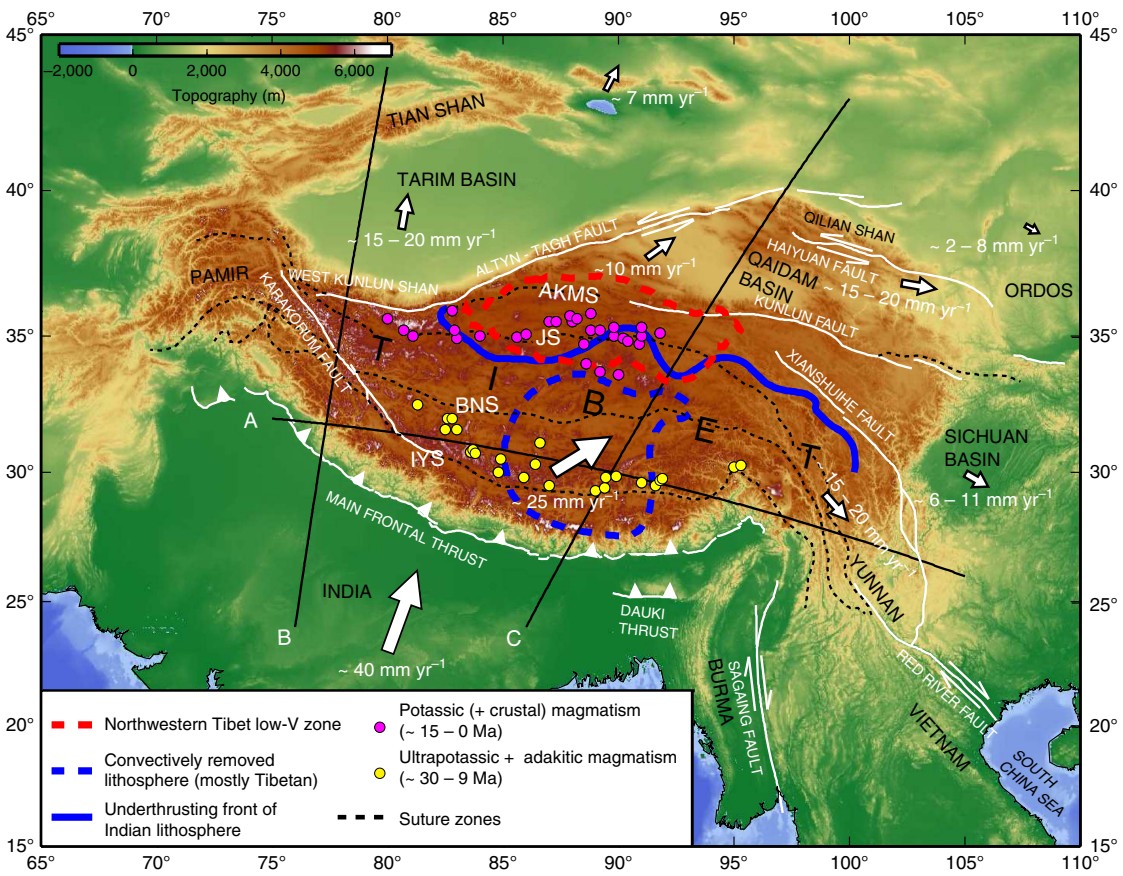

**Figure 1 | Geological map of Tibet and surrounding regions.** Major fault traces (white lines) and suture zones (black dash lines) are obtained from the HimaTibetMap-1.1 data set[69]. Yellow and magenta filled circles mark two different episodes of magmatism distributions[46,53]. White arrows indicate motions of different tectonic units relative to stable Eurasia[2,70]. The thick red dashed line delineates the −4% contour of shear wave speed anomalies ($\delta ln V_S$) at a depth of 80 km beneath Northern Tibet, the thick blue line denotes the 2% contour of $\delta ln V_S$ at a depth of 175 km and the thick blue dashed line represents the 2% contour of $\delta ln V_S$ at a depth of 350 km. All the contour lines are extracted from Fig. 2a–c. The abbreviations of suture zones, IYS, BNS, JS and AKMS are defined in Fig. 4.

tomography, called adjoint tomography, accounts for off-ray path three-dimensional (3-D) sensitivity, takes advantage of multiple seismic phases that sense different parts of the earth, and therefore improves the accuracy of mapping elastic properties of Earth's interior from seismic records[23]. Here we use adjoint tomography (see the 'Methods' section for details), based on a spectral-element method (SEM) and 3-D finite-frequency sensitivity kernels to assimilate full waveform information— including (but not limited to) P and S body waves and Love and Rayleigh surface waves recorded by a wide-aperture dense array— to obtain a new seismic model named EARA2014 (ref. 24). Details of the model construction and its quality assessment are provided in a previous publication[24].

The goal of this study is to interpret observed shear wave speed anomalies in the upper mantle as they relate to the post-collision fate of Indian, Tibetan and Asian mantle lithospheres, and to better understand the connection between lithospheric evolution and surface expressions of plateau uplift and volcanism (Fig. 1). We attribute a T-shaped high wave speed (high-V) structure beneath South-Central Tibet to lithospheric foundering during the Oligocene, which causes surface major uplift and ultrapotassic and adakitic magmatism in Southern Tibet. Overriding the foundering lithosphere, high-V IL underthrusts Tibet as far north as the Jinsha Suture (JS) at present. A narrow low wave speed zone imaged in the uppermost mantle beneath Northern Tibet, consistent with recent potassic volcanism, suggests that surface uplift of Northern Tibet is subject to more recent asthenospheric upwelling and thermal erosion of thickened lithosphere.

## Results

**Mantle tomography.** Mantle shear wave speed anomalies in model EARA2014 vary distinctly across Tibet in both S–N and W–E directions (Figs 2–4). For convenience, we define that Southern and Northern Tibet are separated by the JS, which approximately coincides with the 2% level contour of shear wave speed anomalies at a depth of 175 km (Figs 1 and 2b). We also define that Southern Tibet is divided into three sub-regions from west to east, namely Southwestern, South-Central and South-eastern Tibet, at longitudes of 83°E and 92°E based on the 3-D contour surface of 2% shear wave speed anomalies (Figs 3b and 5a). Two prominent high-V structures are identified in the mantle: a sub-horizontal high-V structure just below the Moho down to a depth of 250 km consistently imaged along the Himalayan arc (Figs 3b,c and 5a) and a T-shaped high-V structure beneath South-Central Tibet extending from 250 km depth to the bottom of the transition zone (Fig. 4e). The strongest low wave speed (low-V) anomalies within Tibet are located in the crust and uppermost mantle in a narrow W–E oriented zone about 200 km wide (Figs 1,2a,4e and 5b,c). This low-V zone follows the JS from longitude 83°E to longitude 95°E and is approximately bounded by the Anymaqen–Kunlun–Muztagh Suture to its north (Fig. 5b,c). The shear wave speeds inside this low-V zone exhibit more than 4% reductions in the uppermost mantle and more than 6% reductions in the crust.

Compared with traditional tomographic methods which heavily rely on 'crustal corrections', adjoint tomography[25] has the advantage of incorporating 3-D crustal structure in the initial model and iteratively updating it with both body- and surface-waveform information, thereby effectively minimizing crustal contamination of the mantle in the final images. The high-V structures at different depths in the upper mantle are well resolved in our study under South-Central Tibet (see the 'Methods' section for details; Supplementary Fig. 1). Resolution tests show that high-V perturbations at both 150 and 400 km depths can be recovered (Supplementary Fig. 1). The upper

250 km of shear wave speeds in the crust and uppermost mantle (Figs 2a,b,3 and 4) especially have improved resolution compared with most of the global or regional tomographic models based on asymptotic methods and 'crustal corrections' (for example, Supplementary Fig. 2, their Figs 7 and 9 in ref. 24). The low-V zone in Northern Tibet is well constrained by both body and surface waves and is laterally more confined along the JS compared to previous results based solely on surface waves[26] or Pn and Sn waves[27–29]. On the other hand, the observed low-V zone in this study is a broadened vertical feature throughout the crust and uppermost mantle due to the lack of very high-frequency waves in our inversion (Supplementary Fig. 1a,b). More robust recovery of the amplitude and depth extent of the low-V anomalies within the crust requires the incorporation of shorter period surface waves ($<20\,s$) and body waves ($<12\,s$), and more regional crustal earthquake data. We will leave discussions of mid-lower crustal flow related low-V anomalies for future full waveform inversion studies and will focus on interpreting the low-V imprint in the uppermost mantle.

## Discussion

We interpret the sub-horizontal high-V structure ($>2\%$ increases) shallower than 250 km in the mantle as IL underthrusting beneath Tibet. The thickness of the under-thrusting IL is between 100 and 150 km based on the 2% level contour of shear wave speed anomalies (Figs 3 and 4). This observation is consistent with a receiver-function study of the interfaces beneath the Indian subcontinent[30], where the derived depths of the lithosphere–asthenosphere boundary vary between 70 and 140 km, and reach up to $\sim170\,km$ beneath the Himalayan region and Moho depths located between 30 and 56 km. Arc-normal cross-sections B and C show that underthrusting IL gently dips northward at an angle of $\sim10°$ (Fig. 4b,e) without visible high-angle subduction in the deeper upper mantle. It is laterally continuous from the Main Frontal Thrust to its northern leading edge, which proceeds beyond the Bangong-Nujiang Suture (BNS) and as far north as the JS (Figs 1,2b and 4, and Supplementary Fig. 2a–d). Our interpreted location of IL's leading edge (Fig. 1), approximately coinciding with the JS, is different from previous interpretations from P-wave models based on traditional tomographic methods. P-wave tomography studies generally map the IL underthrusting/subduction front[19,31–34] very close to (for example, Supplementary Fig. 2f) or to the south of the BNS (for example, Supplementary Fig. 2g,h) in Central Tibet between 87°E and 91°E. Along profile 83°E, a model comparison (Supplementary Fig. 2a,e) shows agreement on that the IL underthrusting front reaches as far north as JS in both model EARA2014 and the global P-wave model used in Replumaz et al.[32] However, their P-wave model[32] reveals a much thicker ($\sim300\,km$ thick) craton-like structure beneath India (Supplementary Fig. 2e), whereas EARA2014 shows a normal thickness of $\sim150\,km$ without invoking underthrusting a very thick continental craton.

Seismicity is distributed along the interpreted upper interface of IL and terminates at depths shallower than $\sim100\,km$ (Fig. 4c,f). Except along profile B, one earthquake at a depth of 140 km located by the EHB catalogue[35] (event ID: 12477266) occurs in the vicinity of the interpreted IL upper interface (Fig. 4c). There is a large gap between this deep earthquake and shallower crustal seismicity along the interface of IL. Therefore, it probably belongs to the southward subducting Pamir slab, annotated as Asian lithosphere (AL) (Fig. 4c). It is possible that the IL upper interface is steeper under Himalayan blocks between the Main Frontal Thrust and the IYS[36] and flattens further

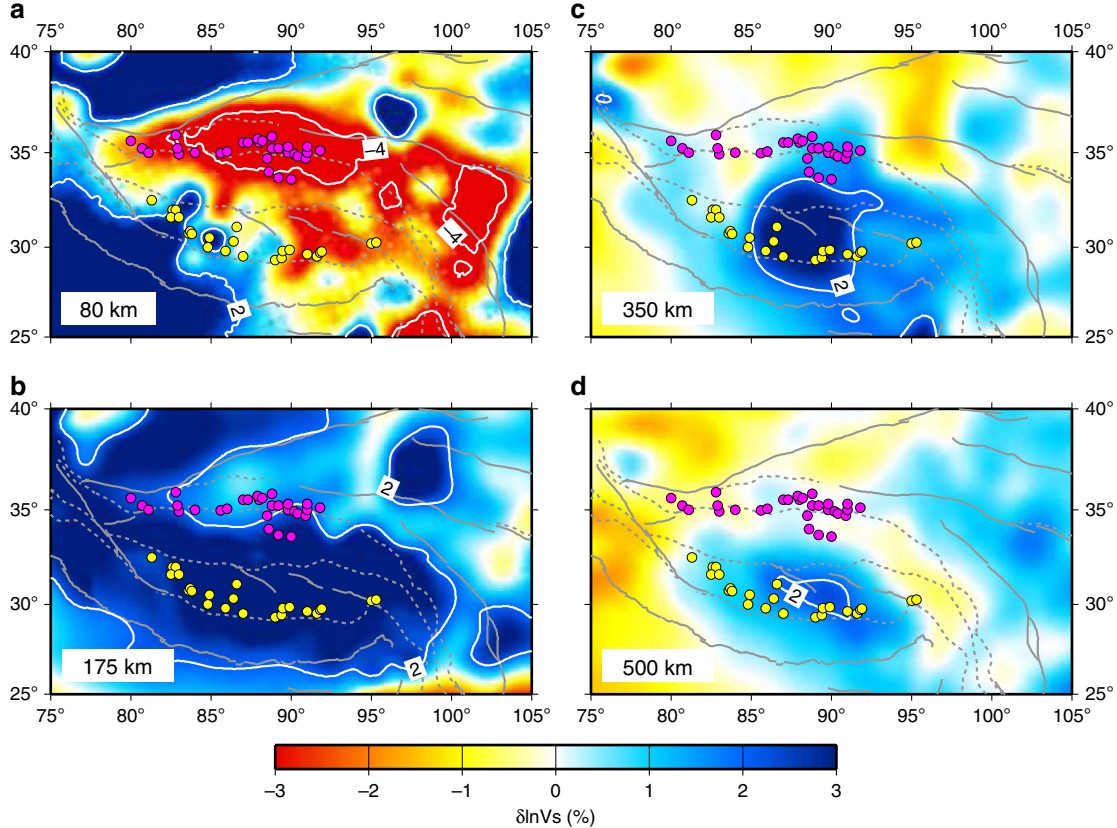

**Figure 2 | Shear wave speed anomalies at different depths.** Map views at (**a**) 80 km, (**b**) 175 km, (**c**) 350 km and (**d**) 500 km depth. Major fault traces and suture zones are delineated in grey solid lines and grey dashed lines respectively. Yellow and magenta filled circles mark two different episodes of magmatism distribution (Fig. 1). White lines delineate $-4$ and 2% contours of $\delta \ln V_S$.

north[37], but such details cannot be resolved in this study due to resolution limits.

No large-scale low-V anomalies are discernable within the underthrusting IL, which does not support the hypotheses of IL being fragmented due to delamination and asthenosphere upwelling[20,21]. Low-V anomalies beneath Southern Tibet are only visible at depths shallower than 150 km. Such low-V anomalies (more than 2% reductions) imply possible partial melting. The low-V zones located at crust and uppermost mantle depths do not have a visible connection to any deeper mantle low-V zones. This suggests that partial melting is not a mantle driven process, but instead a crustal process either related to shear heating generated in ductile shear zones near the India-Himalaya lithospheric interface[38] or to radioactive heating within the crust[39].

Lateral heterogeneities do exist within the interpreted IL in the arc-parallel direction, where beneath the Southern Tibet rift region (83°E–95°E) ~200 km wide strongly high-V zones (more than 4% increases) alternate with ~100 km wide relatively weakly high-V zones (3–4% increases) (Fig. 3c). This suggests that underthrusting IL is probably intact, with local weaker zones representing either pre-existing, that is, before initial subduction, structures or locally modified regions due to melt and/or volatile injection after subduction. Absolute values of shear wave speeds in the underthrusting region range from 4.7 to 4.8 km s$^{-1}$ (Figs 3c and 4c,f, and Supplementary Figs 3 and 4), comparable to those of the North American craton and much higher than in active tectonic regions ($<4.5$ km s$^{-1}$) in the uppermost mantle[40]. If the underthrusting IL can be treated as the root of the present-day TL, then the lithospheric structure of Tibet resembles that of Archaean and Proterozoic cratons, except with a hotter and thicker crust at present, which may be gradually eroded at the

top and become more similar in terms of crustal thickness to Archaean and Proterozoic cratons[39].

Underlying underthrusting IL, the T-shaped high-V structure beneath South-Central Tibet has a less obvious origin (Figs 4e and 6d). Its top part is located above the transition zone with a height of about 150 km and an arc-normal width of ~750 km spanning from latitude 28°N (south of the IYS) to latitude 33°N (north of the BNS). Its bottom part resides in the transition zone with a height of ~250 km and an arc-normal width of ~200 km, situated between the IYS and BNS. In contrast to a narrow high-V structure observed from a depth of 250 km to the top of the transition zone beneath Western Pacific regions (for example, the Japan and Izu-Bonin convergent margins), which is associated with abundant deep-focus seismicity and interpreted as subducting oceanic lithosphere[24], the deep mantle high-V structure beneath South-Central Tibet is a much broader feature and completely lacks seismicity. Such striking differences indicate that the T-shaped high-V structure is unlikely subducting oceanic lithosphere and the portion of Indian oceanic lithosphere probably already sank into the lower mantle[16,32,41,42]. This argument is further bolstered by a simple estimation of the total budget of consumed continental lithosphere after complete subduction of Indian oceanic lithosphere. The total budget of continental lithosphere, possibly from Indian, Tibetan, or Asian continental blocks, that entered the mantle since continental collision is conservatively estimated at ~2,250 km in length, given an average convergence rate of 50 mm per year since 45 Ma[1]. However, the observed IL overriding the T-shaped high-V structure is ~750 km in length (Fig. 6d), which accounts for only one-third of the total budget and leaves the remaining ~1,500 km unaccounted for. If the imaged T-shaped high-V

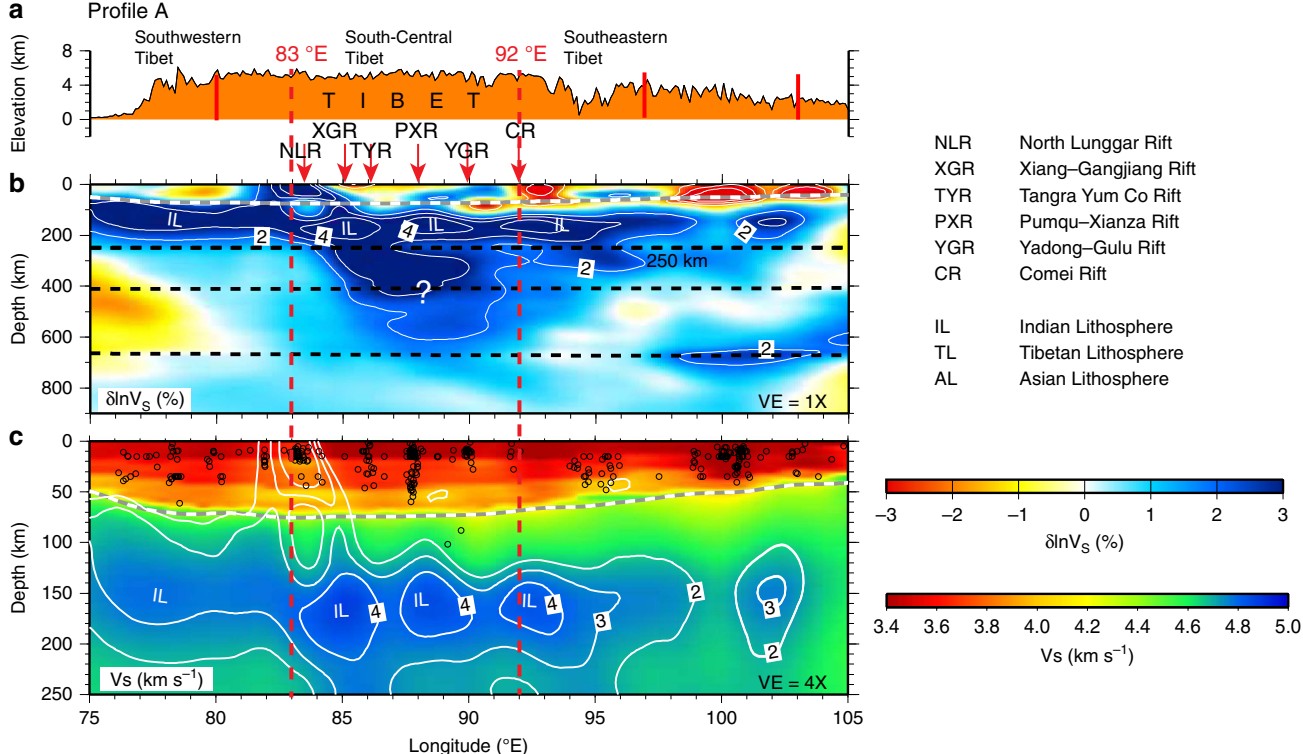

**Figure 3 | Cross-sections showing surface elevations and seismic structures along an arc-parallel profile.** (**a**) The surface elevations, (**b**) shear wave speed anomalies ($\delta \ln V_S$) and (**c**) shear wave speeds ($V_S$) along profile A (Fig. 1). In **b**, black dashed lines mark a depth of 250 km and the 410 and 660 discontinuities, and white lines represent $\delta \ln V_S$ contour levels from $-4$ to $-2\%$ and from 2 to 4% at 1% intervals. In **c**, black circles denote the seismicity, $V_S$ are plotted with $4\times$ of vertical exaggeration (VE $= 4\times$), and white lines represent $\delta \ln V_S$ contour levels from 2 to 4% at 1% intervals extracted from **b**. In **b** and **c**, grey dashed line delineates the Moho from CRUST2.0 (ref. 25). The abbreviations of rift zones, NLR, XGR, TYR, PXR, YGR and CR, and the different lithospheric block, IL, TL and AL are defined in the text block on the right.

structure is interpreted as a foundering continental mantle lithosphere, that is, the majority of the thickened continental mantle lithosphere detached at the bottom but with some part of the top portion still left attached to the crust above, then unwrapping the area of the imaged anomaly to a 120 km[30] thick pre-collision lithosphere gives a length estimate of about 1,354 km (equation 1 and Fig. 6d),

$$\text{Length} = (750 \times 150 + 200 \times 250)/120 = 1354 \,(\text{km}) \qquad (1)$$

which makes up the majority of the missing post-collision continental lithosphere. Therefore we argue that the T-shaped high-V structure is of continental lithospheric origin.

It is still yet to be determined if the detached T-shaped lithosphere is derived from Indian, Tibetan, or Asian continental blocks, because all three continental blocks have the possibility of entering and remaining in the upper mantle through different processes, such as subduction followed by slab breakoff[16], or lithospheric thickening followed by convective removal[43], that is, foundering in this discussion. If AL subducts southwards under Tibet and later breaks off, a south dipping slab structure would be expected under Northern Tibet from either the Tarim or Qaidam Basins. A previous receiver-function study images a prominent south-dipping interface down to 250 km beneath northern Tibet and interprets it as the top of south dipping AL[44]. However, there is no compatible seismic tomographic evidence showing positive wave speed jumps downward across the imaged interface (for example, their Fig. 4 in a previous P-wave tomography study[42]). Alternatively, the receiver-function interpreted south dipping AL interface[44,45] can be reconciled with the strong wave speed contrast between our interpreted weakly high-V TL and the strongly low-V zone above (Fig. 4e),

which we speculate as an internal interface within TL. Moreover, consistent with previous tomographic results[20,21], no obvious evidence of south dipping AL under Northern Tibet is shown in the arc-normal cross-section (Fig. 4e), because weakly high-V anomalies ($<1\%$ increase) interpreted as TL are significantly weaker than strongly high-V anomalies (2 to 5% increases) interpreted as AL. In the W–E oriented cross section along latitude 36°N, AL is also outlined by strong high-V anomalies of 2 to 5% down to a depth of at least 250 km under the Qilian Shan fold-thrust belt and is seismically discernible from TL that has weakly high-V anomalies of less than 1% (Supplementary Fig. 5). Although our observation does not support the model involving AL southward subduction leading to growth of crustal accretionary wedges[12,13], it is possible that AL subducts eastward at a dip angle of $\sim25°$ from the Qaidam Basin and contributed to the high-elevation of the Qilian Shan fold-thrust belt (Supplementary Fig. 5).

Therefore, continental lithosphere more likely foundered from IL or TL or both, although their relative contributions depend on the pre-collision thickness and strength of both lithospheric blocks. As TL is considered to be hotter and, as result, most likely to be rheologically weaker than colder IL[46], we speculate that Tibetan mantle lithosphere is more prone to thicken along with the crust right after continental collision starts (Fig. 6a). The colder and stronger Indian mantle lithosphere is more likely to undergo underthrusting without significant internal deformation. Continued penetration of IL is resisted by thickened TL and is likely to be limited to a few hundred kilometers of distance in the arc-normal direction (Fig. 6a). Owing to the Rayleigh–Taylor instability[43,47], the viscous lower part of thickened Tibetan mantle lithosphere can initially 'drip'

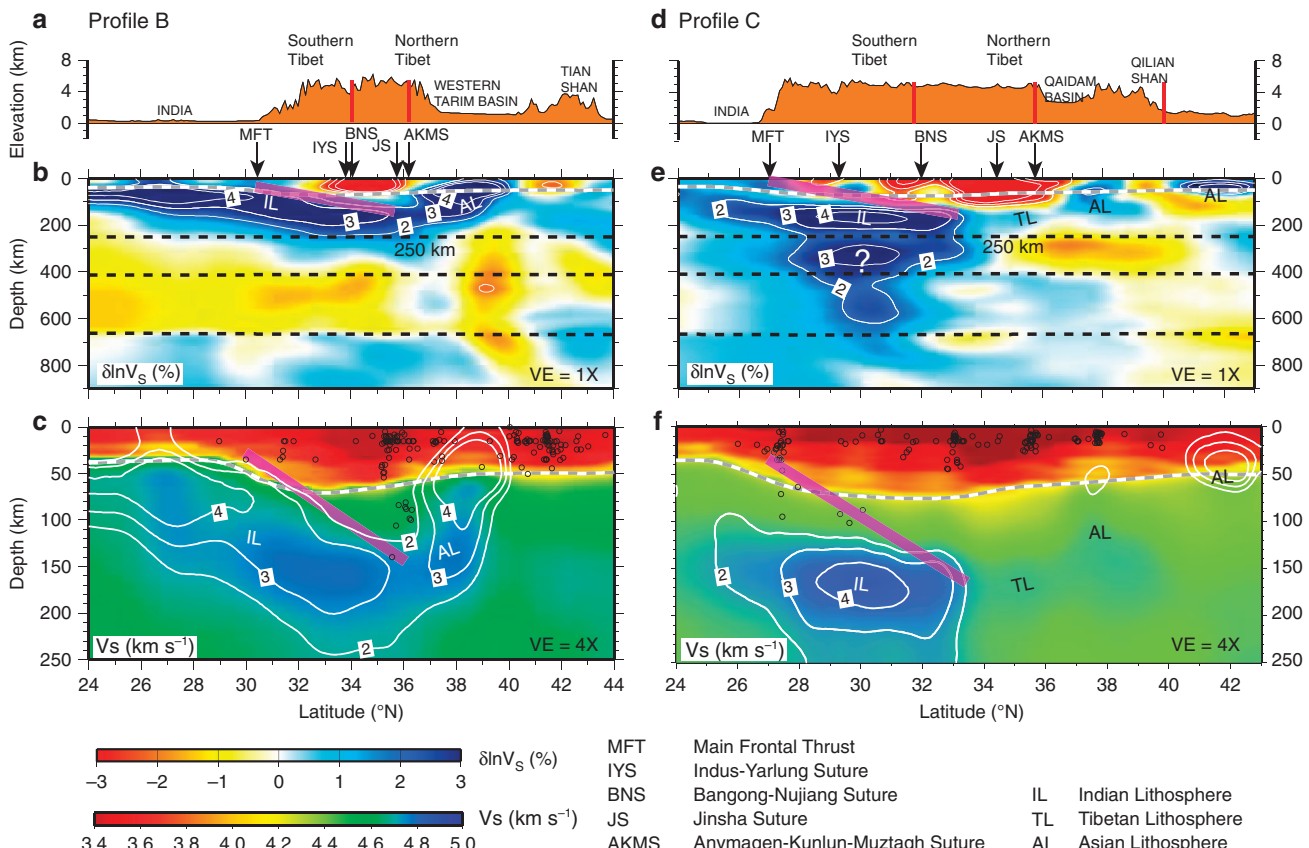

**Figure 4 | Cross-sections showing surface elevations and seismic structures along arc-normal profiles.** (**a**) The surface elevations, (**b**) shear wave speed anomalies ($\delta \ln V_S$) and (**c**) shear wave speeds ($V_S$) along profile B (Fig. 1). (**d**) The surface elevations, (**e**) $\delta \ln V_S$ and (**f**) $V_S$ along profile C (Fig. 1). In **a** and **d**, vertical red bars indicate major fault zones. In **b** and **e**, black arrows mark the Main Frontal Thrust (MFT) and suture zones (IYS, BNS, JS and AKMS) and white lines represent $\delta \ln V_S$ contour levels from $-4$ to $-2\%$ and from 2 to 4% at 1% intervals. In **c** and **f**, black circles denote the seismicity, $V_S$ are plotted with $4\times$ of vertical exaggeration (VE $= 4\times$), black dashed lines mark a depth of 250 km and the 410 and 660 discontinuities, and white lines represent $\delta \ln V_S$ contour levels from 2 to 4% at 1% intervals extracted from **b** and **e**, respectively. In **b**,**c**,**e** and **f**, grey dashed lines delineate the Moho from CRUST2.0 (ref. 25) and thick magenta lines represent the interpreted upper interface of underplated IL with a dip angle of 10°.

on a relatively small scale ($\sim 200$ km wide), followed by breakoff of the more rigid upper part ($\sim 750$ km wide) of IL and TL accommodated by faults or other weak zones[47].

The timing of lithospheric foundering beneath Southern Tibet can be constrained by the timing of ultrapotassic and adakitic magmatism that initiates at about $\sim 30$ Ma and lasts until $\sim 9$ Ma (Figs 1 and 6a–c)[46]. Post-collision adakitic magmatism suggests the occurrence of thickening of TL and subsequent lithospheric root foundering. Lithospheric foundering significantly thinned Southern TL that was thickened before 30 Ma due to continental collision. The loss of lithospheric root can drive surface uplift during the Oligocene[48] and observed ultrapotassic and adakitic magmatism, fueled by the ascent of asthenospheric return flow. The continued sinking of foundering lithosphere in the upper mantle can also generate lateral pressure gradients in viscous asthenosphere that can drive shear traction at the base of overlying mantle lithosphere[49]. This shear traction drives northward underthrusting of IL and thickening of remaining TL in the north (Fig. 6b–d). The northward advance of underthrusting IL gradually shuts off sources of heat and melting and causes waning of ultrapotassic and adakitic magmatism in Southern Tibet[46].

We conclude that the leading edge of IL has moved northwards over an arc-normal distance of about 750 km (Figs 4e and 6d) since the acceleration of underthrusting at $\sim 25$ Ma (Fig. 6b), when the lower part of the pre-thickened lithosphere becomes

detached completely. This interpretation gives an estimated average underthrusting rate of about 30 mm per year in the past 25 million years. It is slightly higher than the current ongoing convergence rate of $\sim 20$ mm per year between India and the IYS, but remains a reasonable estimate as convergence is expected to have slowed down due to resistance associated with thickened lithosphere[50].

Northern TL is probably being heated by asthenospheric upwelling. The S-N contrast in shear wave speed perturbations in our model (Supplementary Fig. 3) is compatible with other results independent from seismic tomography. Based on our observed 3% of S–N $V_S$ difference and a relation between $V_S$ perturbation and temperature of $1.3 \pm 0.30\%$ per 100 K at 200 km[51], TL under Northern Tibet is estimated to be 200–300 K warmer than underthrusting IL beneath Southern Tibet. Such temperature difference in the uppermost mantle agrees with heat flow modelling[52]. The S–N difference of the average shear wave speed between the surface and 410 km depth (about 3% along profile D; Supplementary Fig. 3) is also consistent with receiver function observations of the 410- and 660-discontinuities being parallel and relatively depressed in the south[44]. Thickened Northern TL may be gradually eroded or thermally modified by hot asthenospheric upwelling (Fig. 6c,d). The good spatial correlation between the strongly low-V zone in the uppermost mantle and more recent potassic magmatism in Northern Tibet ($\sim 15$–0 Ma)[46,53] (Figs 1,5b,c and 6c,d) further support the

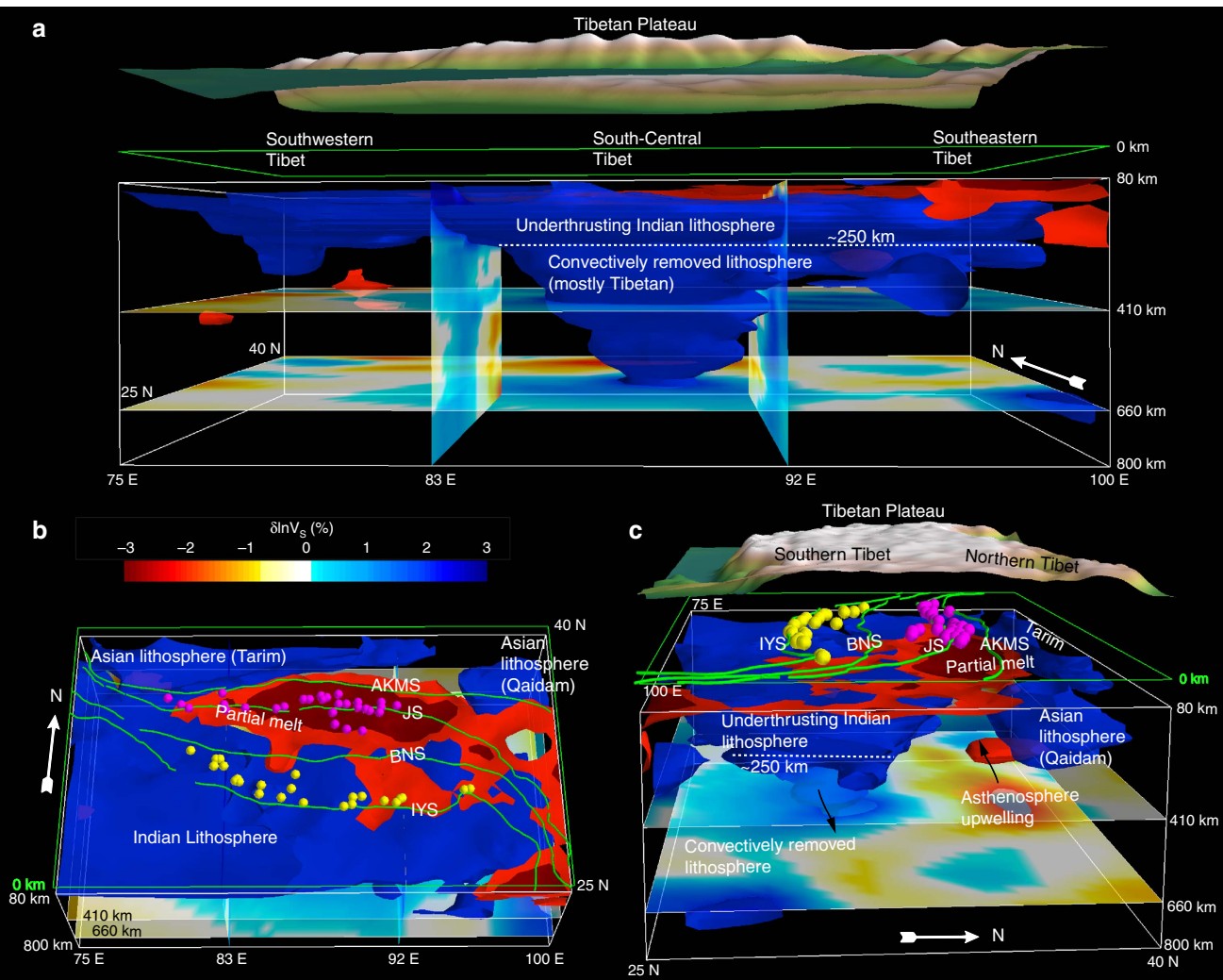

**Figure 5 | Three-dimensional visualization of the shear wave speed structure under the Tibetan Plateau.** The $-4\%$ (dark red), $-2\%$ (red) and 2% (blue) isosurfaces of $\delta \ln V_S$ are rendered from EARA2014 (ref. 24). Green lines mark the four suture zones IYS, BNS, JS, and AKMS. For reference, four planes showing variations of $\delta \ln V_S$ are cut at depths of 410 and 660 km, and along longitudes 83°E and 92°E. (**a**) The geometry of Indian (blue) and Asian (blue) lithospheres and the distribution of possible partial melt (dark red) are viewed upward from the south, (**b**) downward from the south and (**c**) from the east.

hypothesis of asthenospheric upwelling. The low-V anomalies are, however, limited to the uppermost mantle ($<125$ km) overlying weakly high-V TL that extends down to a depth of $\sim 200$ km. Contrary to more dramatic lithospheric foundering and thinning during the Oligocene in Southern Tibet, Northern TL more likely to be experienced 'diffused' root removal or thermal modification and is still largely intact. Thermal modification can lead to a more buoyant lithospheric mantle that isostatically supports uplift of Northern Tibet.

Our results are consistent with the following conceptual model. Distributed thickening of TL and underthrusting of IL accommodate the bulk of mantle lithosphere convergence since India-Eurasia collision. Convergence leads to shortening and thickening of TL, including both crust and mantle. Subsequent foundering of thickened lithosphere during the Oligocene contributed to the rise of Southern Tibet. The foundering lithosphere is continental in origin and, as a result, is less negatively buoyant than oceanic lithosphere. This can lead to a long residence time ($\sim 30$ Ma) of foundering continental lithosphere in the upper mantle. In addition, the 660-discontinuity can act as rheological and density barrier preventing foundering continental lithosphere from sinking into the lower mantle (Fig. 6d).

Different from pure subduction settings, where lithosphere subducts without much thickening, the India-Eurasia continental collision zone involves thickening of the continental mantle lithosphere (TL) and low-angle underthrusting of stronger IL. Deformation and thickening on the Tibetan side is not confined to the crust and is more vertically distributed throughout the entire column of crustal and mantle lithosphere. Wholesale thickening of TL can initiate a Rayleigh–Taylor instability and subsequent foundering (convective removal) of the lithospheric root. Convective removal and associated litho-spheric foundering creates an additional plate driving force, an asthenospheric drag force, resulting in continued thrusting of IL under Tibet. This is different from the principal driving force of plate tectonics at oceanic subduction zones, which is created by negative buoyancy of dense oceanic mantle lithosphere. The direct impact of such convective removal is a more pulsed surface uplift of Southern Tibet over a time scale of $<10$ m.y. ($\sim 30$–25 Ma) rather than over the entire 45 m.y. of India-Eurasia collision.

If northward penetration of IL slows down exponentially due to resistance from viscous mantle lithosphere[50], it is likely to be that India-Tibet convergence will terminate by the time IL occupies the entire uppermost mantle underneath Tibet.

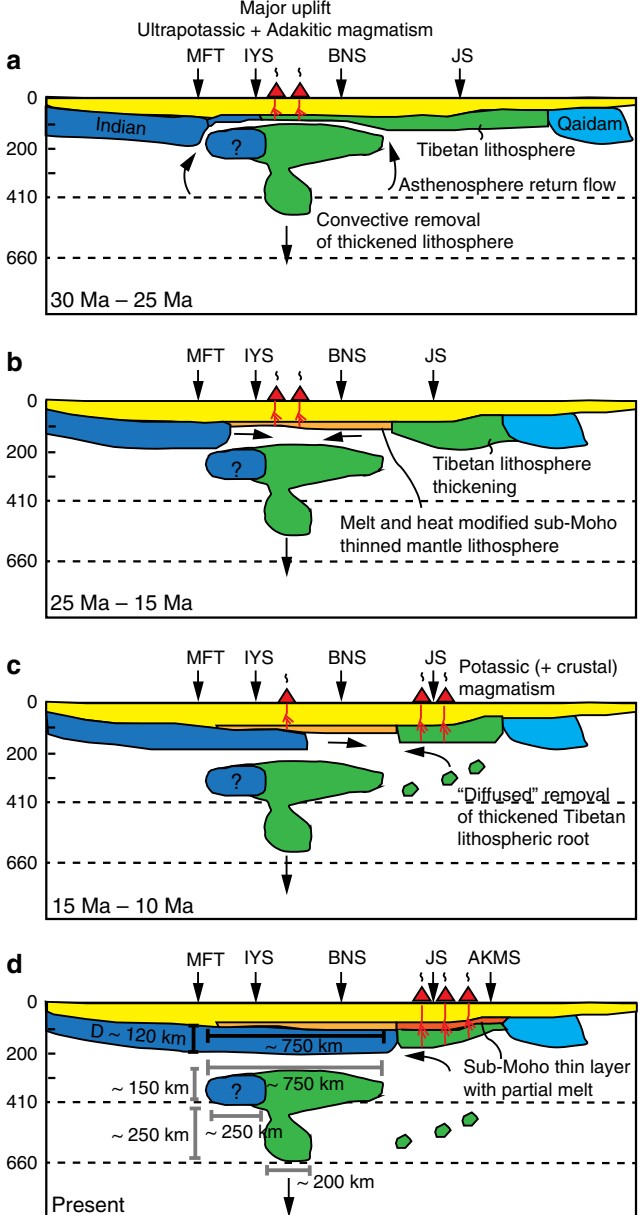

**Figure 6 | Inferred tectonic evolution of Tibet.** The interpretation is based on the seismic image along profile C (Fig. 4e) as well as previous studies on magmatism[46,53]. (**a**) Between 30 Ma and 25 Ma: following lithospheric thickening due to continental collision, convective instability triggers removal of a lithosphere root and surface uplift. Asthenospheric return flow initiates ultrapotassic and adakitic volcanism in Southern Tibet. (**b**) Between 25 Ma and 15 Ma: magmatism persists in Southern Tibet while partial melt and heat modify the remaining thin uppermost mantle lithosphere. (**c**) Between 15 Ma and 10 Ma: further northward underthrusting of IL gradually shuts down the heat source of magmatism in Southern Tibet. (**d**) Present: Southern Tibet is completely underthrusted by IL up to the south of the JS. Magmatism in Northern Tibet is still an ongoing process.

The strength and buoyancy of Indian continental lithosphere might keep it in place beneath Tibet for a substantial amount of time, possibly long enough to be considered the root of a stable craton. This might provide a mechanism for the formation of a modern craton in the Tibet-Himalaya continental collision margin, consistent with geodynamic modeling[54] and similar to a previously proposed mechanism of craton formation through underthrusting and imbrication of oceanic lithosphere[55],

however, through under-accretion of Indian continental lithosphere instead.

## Methods

**Adjoint tomography and model construction.** Seismic images of Tibet and its surrounding regions are rendered from East Asia Radially Anisotropic Model (EARA2014)[24]. This structural model is developed using adjoint tomography, assimilating 1.7 million frequency-dependent traveltime measurements from waveforms of 227 earthquakes recorded by 1,869 stations in East Asia. The majority of stations are from the CEArray[56] densely covering China. Tibet has complementary station coverage from INDEPTH (International Deep Profiling of Tibet and the Himalaya) IV two-dimensional broadband deployment and other regional and global arrays. Adjoint tomography in this application utilizes a highly accurate SEM to simulate 3-D seismic wave propagation[57,58] and to calculate finite-frequency sensitivity kernels for iterative tomographic inversion[59–61]. Technical details of model construction are described in a previous publication[24]. Here we briefly summarize the data and method. Our initial model consists of a 3-D global radially anisotropic mantle model S362ANI[62] and a 3-D crustal model Crust2.0[25]. Initial earthquake source parameters are described by the centroid moment-tensor (CMT) solution[63]. A total of 227 earthquakes (Mw = 5–7) with good signal-to-noise-ratio records are selected from the global CMT solution database. Source parameters are reinverted using CMT3D inversion method[64] with synthetic waveforms simulated in the initial 3-D structural model on global scale. Seismic waveforms from five high-quality global and regional seismic networks (IU, II, G, GE and IC) are used in the source inversion to insure good global azimuthal coverage. Observed and synthetic waveforms are bandpass filtered in three complementary period bands, namely, from 30 to 60 s, 50 to 100 s and 80 to 150 s. Body wave misfits in the period range 30–60 s and body wave and surface wave misfits in 50–100 s and 80–150 s passbands are used in the source inversions. After the source inversions, the subsequent iterative structural inversion takes place in a wave simulation volume described as a 80° by 80° spherical chunk laterally centered on China and vertically spanning from the surface to Earth's core. All the used earthquakes and stations are contained in the model simulation volume. Our regional 3-D models have an isotropic parameterization in the crust and in the mantle below the transition zone, and a radially anisotropic parameterization between the Moho and the 660 km discontinuity. The SEM mesh incorporates a 4-min topography model created by subsampling and smoothing ETOPO-2 (ref. 65), as well as undulations of the Moho[25], and the 410 and 660 km discontinuities[62]. We updated the 3-D regional structure based on finite-frequency kernels with fixed source parameters. Our data set for structural inversion consists of three-component waveforms recorded by 1,869 stations from F-net, CEArray[56], NECESSArray, INDEPTH IV Array and other regional and global seismic Networks. The regional structural model is parameterized on the SEM Gauss–Lobatto–Legendre integration points, which have an 8 km lateral spacing and a vertical spacing of <5 km in the crust, and a 16 km lateral spacing and an average vertical spacing of ~10 km in the upper mantle. Synthetic seismograms for the initial 3-D model and subsequent updated models were calculated for all stations. Measurement windows are selected in three passbands, namely 15–40 s, 30–60 s and 50–100 s for the first 12 iterations. In subsequent iterations we lowered the lower bounds of these passbands to 12, 20 and 40 s, respectively. Selecting measurement windows is accomplished based on FLEXWIN[66], an algorithm to automatically pick measurement windows in vertical, radial and tangential component seismograms by comparing observed and synthetic seismograms. Frequency-dependent traveltime misfits are measured within the chosen windows. Adjoint sources are constructed using traveltime misfit measurements for all picked phases, for example, body wave phases (direct P and S, pP, sP, sS, pS, PP and SS) and surface waves (Rayleigh and Love). The adjoint sources assimilate the misfit as simultaneous fictitious sources, and the interaction of the resulting adjoint wavefield with the regular forward wavefield forms the event kernels. All event kernels are summed to obtain the gradient or Fréchet derivative, which is preconditioned and smoothed for a conjugate gradient model update. The optimal step length for the model update is chosen based on a line search. The updated model is used as the starting model for the next iteration of further structural refinement. The same procedure is repeated until no significant reduction in misfit is observed, in our case after 20 iterations.

**Resolution test.** Model quality of EARA2014 is extensively assessed by examining waveform misfit reductions, establishing regions with reasonably good data coverage, comparing with previous tomographic models, performing resolution tests at several locations of interest, and an inversion with a different initial model. For details of model quality assessment please refer to previous publication[24]. Here we focus on 'point-spread function' (PSF) resolution tests targeting the uppermost mantle and mid-upper mantle high-V Indian lithospheric structure beneath South-Central Tibet. The PSF test evaluates the resolution of a particular point of interest in the model by the degree of 'blurring' of a perturbation located at that point, and by revealing the tradeoff with other model parameters[67,68]. We placed a spherical anomaly represented by 3-D Gaussian functions centered at two different depths, 150 km (uppermost mantle) and 400 km (mid-upper mantle) beneath South-Central Tibet, with a 120 km radius and a maximum of 4%

perturbation in $V_{SV}$ (Supplementary Fig. 1a,c). Although there is certain degree of smearing (Supplementary Fig. 1b,d), $V_{SV}$ PSFs at both depths recover the main features of the perturbations. On the other hand, our resolution tests (Supplementary Fig. 1) also suggest that the T-shaped feature at deeper depth (250 km and deeper) might be artificially mapped about 50 km upwards (Supplementary Fig. 1c,d) due to the vertical smearing effect of body wave resolution at this depth range.

**Model analysis.** Tibetan upper mantle has relatively higher wave speeds compared with the rest of East Asia in model EARA2014 (ref. 24). The regional mean of shear wave speeds at each depth is calculated for a region spanning from 65°E to 120°E and 20°N to 45°N (Supplementary Fig. 3). In the seismic images (Figs 2–4 and Supplementary Fig. 5) and 3-D visualizations (Fig. 5), the regional mean at each depth has been removed and converted to percentage perturbations to emphasize wave speed variations in Tibet and surrounding regions.

**Data availability.** Digital file of model EARA2014 in the study region of this manuscript is available upon request to the corresponding author.

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

## Acknowledgements

We thank the various networks that contributed data (F-net, CEArray, NECESSArray, INDEPTH IV Array, IRIS/IDA and other regional and global seismic networks), as well as the Rice Research Computing Support Group. The majority of waveform data were provided by the China Seismic Array Data Management Center at the Institute of Geophysics, China Earthquake Administration. We also thank Dr Anne Replumaz and another anonymous reviewer for their constructive comments and suggestions, which significantly improved the quality of this paper. This research was supported by NSF grant 1345096. This work used the Extreme Science and Engineering Discovery Environment (XSEDE), which is supported by NSF grant ACI-1053575. The open source spectral-element software package SPECFEM3D_GLOBE, the seismic measurement software package FLEXWIN and the moment-tensor inversion package CMT3D used for this article are freely available for download via the Computational Infrastructure for Geodynamics (CIG; geodynamics.org).

## Author contributions

M.C. conducted the adjoint tomography and took the lead in writing the manuscript. M.C. and F.N. contributed to the data acquisition. J.T. contributed to the theory of adjoint tomography. M.C., F.N., A.L., C.-T.A.L., W.C. and J.R. contributed to the model interpretation. All authors contributed to the manuscript writing.

## Additional information

**Competing interests:** The authors declare no competing financial interests.

