## [Peer Review File · Nature Communications]

Reviewers' Comments:

Reviewer #1 (Remarks to the Author)

This paper presents the results of a new tomographic model of Tibet (EARA2014), using a new method, called adjoint tomography, taking advantage of multiple seismic phases, as P and S body waves and Love and Rayleigh surface waves, recorded by a wide-aperture dense array. Compared to traditional tomographic methods which heavily rely on "crustal corrections", adjoint tomography has the advantage of incorporating 3-D crustal structure in the initial model and iteratively updating it with both body and surface waveform information, thereby effectively minimizing crustal contamination of the mantle in the final images.

The new model seems valuable, improving the resolution of the tomographic images beneath Tibet. But I am not a specialist and I cannot judge the pertinence of the method used, or the choice of parameters in the modeling.

In the last decades, global tomography became a very powerful tool to explore the roots of the mountain ranges. Having a new tomographic model with a better resolution could be of great interest for the community. For example a new model to map precisely the underthrusting front of Indian lithosphere, interpreted in this paper approximately along the Jinsha suture (Line 85), but interpret to be along the Bangong suture from body waves tomography (e.g. Replumaz et al., 2013), is of great interest.

But unfortunately I learnt more by reading the supplementary material than the paper itself. In the paper, the figures are too small or too complex to see the data !

Figure 1 is too complex, many contours from tomography, petrological data, faults, sutures, a lot !

Figure 2: too small, with the green circles masking the crustal part of the section, many rifts indicated by red arrows but impossible to see if they can be seen on the tomographic sections. It is said in the text that "Seismicity is distributed along the interpreted upper interface of Indian lithosphere and terminates at depths shallower than 150 km" but it is impossible to see it on the figure. Even on figure S5 it is not clear. Similarly the "~200 km wide strongly high-V zones (more than 4% increases) alternate with ~100 km wide relatively weakly high-V zones (3%-4% increases)" are also not clearly visible on the figure, which is very frustrating.

Figure 3: 3D view with so much text on it (probably because the figure is not clear !), the vertical and horizontal sections are not clearly visible.

On the contrary, in the supplementary material, all the figures are clear, at the right size, with separated figures for separated data, which make the interpretations of the authors clearer and more convincing.

The other main point with the paper is that the authors use a scenario already published (Chung et al., 2005), but their new data add more problems than more constraints to this model: as you show no sinking of the detached lithosphere, it suggests that there is no convective instability of the Tibetan lithosphere, which does not supported the published scenario. In the budget of the indian lithosphere related to this scenario about half of India is missing. The authors should evoke the deep tomographic anomalies (van der Voo et al., 1999; Replumaz et al., 2010, 2013). In this scenario, it also implies a convergence rate for India higher than estimated, considering that the convergence rate is expected to have slowed down in the early collision time (Guillot et al., 2008).

The other key point is the subduction Asian lithosphere visible on seismic profiles (e.g. Kind et al., 2002). Using global tomography to see such subduction is in fact not obvious as the asian lithosphere amplitude anomaly is weaker than the amplitude related to indian lithosphere. Nevertheless such subduction is visible in the Pwaves tomography, as a weak anomaly (Replumaz et al., 2013). You have to discuss why you do not see it in your model. You can argue that you don't see the Asian subduction because your resolution is not high enough, you cannot say that it

does not exist.

To conclude, I really think that the new tomographic model is highly valuable for the community, but I found the paper as it is presented not enough focus on the new data. The authors should first clearly describe the new model, using the clear figures of the supplementary material. I suggest to compare more deeply their results with existing models, in particular Pwave global tomographic models, maybe doing more synthetic tests to show the resolution of their model. And maybe less focus on a model already published, and not very well constrain.

Anne Replumaz

Specific comments

Line 85: "underthrusting front of Indian lithosphere" is a result of your new model to discuss in detail, not a fact ! You have to compare with Pwaves global tomographic models (e.g. Replumaz et al, 2014)

Line 86: "Southern Tibet is divided into three sub-regions from west to east" I don't find this description pertinent, there is no division of Tibet, there is a deep positive anomaly beneath the central part of Tibet.

Line 110: also visible on the synthetic tests (S1).

Line 118: again the northern extent of the indian lithosphere is something to discuss !

Line 123: "Seismicity is distributed along the interpreted upper interface of Indian lithosphere and terminates at depths shallower than 150 km" impossible to see something with a so small figure ! even on figure S5 it is not clear...

Line 131: reasoning too fast, it implies that it is a crustal process, not a mantle driven process.

Line 135: "~200 km wide strongly high-V zones (more than 4% increases) alternate with ~100 km wide relatively weakly high-V zones (3%–4% increases) (Fig. 2a)." impossible to see, which is very frustrating...

Line 145: "lithospheric structure of Tibet resembles that of Archaean and Proterozoic cratons, except with a hotter and thicker crust at present" i really don't think that the comparison is pertinent ! it has to be removed.

Line 149: "the T-shaped high-V structure beneath South-Central Tibet has a less obvious origin" : i agree, not obvious at all... I am not convinced by your interpretation (see discussion), you have to compare with the Pwave global tomography, showing a very different geometry, leading to a different interpretation.

Line 156: "Japan and Izu-Bonin convergent margins is associated with abundant deep-focus seismicity and interpreted as subducting oceanic lithosphere, the high-V structure beneath South-Central Tibet is a much broader feature and completely lacks seismicity." yes for sure it is not an oceanic subduction, you should also remove this not pertinent comparison.

Line 159: "T-shaped high-V structure is unlikely subducting oceanic lithosphere and the portion of Indian oceanic lithosphere probably already sank into the lower mantle." Yes it has been shown by interpreting the deep tomographic anomalies (van der Voo et al., 1999; Replumaz et al., 2010), it should be evoked here, and more taken into account by the authors in their reasoning for the lithosphere budget (Replumaz et al., 2013).

Line 171: "length estimate of about 1,354 km (supplementary text S3)" yes but you don't interpret it as indian but Tibetan lithosphere, so it is still missing half of India ! there is no length estimation in S3 but figure 4.

Line 181: "no obvious evidence of south dipping Asian lithosphere under Northern Tibet is shown in the arc-normal cross section (Fig. 2c and Supplementary Text S3), which is consistent with previous tomographic results." It is really a key point here: the Asian lithosphere subduction is not obvious because the asian lithosphere amplitude anomaly is weaker than the amplitude related to indian lithosphere. Nevertheless Kind et al (2002) showed the subduction of the asian lithosphere. Such subduction is visible in the Pwaves tomography, as a weak anomaly (Replumaz et al., 2013). You have to discuss why you do not see it in your model. You can argue that you don't see the Asian subduction because your resolution is not high enough, you cannot say that it does not exist.

Line 186 : "the Tibetan lithosphere is considered to be hotter and, as result, most likely rheologically weaker than the colder Indian lithosphere, we speculate that the Tibetan mantle lithosphere is more prone to thicken" reasoning not clear, it is hotter now as it has thickened because it was weaker.

Line 193: "Due to the Rayleigh-Taylor instability, the viscous lower part of thickened Tibetan mantle lithosphere can initially drip" It is another key point in your reasoning here: how do you explain that it is dense enough to drip but not to sink in the mantle and stay just below the Indian lithosphere for 30 Ma ?

Line 199: there is also potassic magmatism between 40 and 30 Ma, much wider than the ultra-potassic you show, you cannot choose between magmatism pulse like that.

Line 204: maybe, but the plateau is much wider than the deep anomaly, so it cannot explain the formation of the plateau.

Line 206: "The continued sinking of foundering lithosphere in the upper mantle" : it is one of the main problems of your model: the sinking is very limited. How could you have a convective instability which reheated the crust, without sinking down to the lower mantle ?

Line 218: "It is slightly higher than the current ongoing convergence rate of ~20mm/yr between India and Indus-Yarlung Suture but remains a reasonable estimate as the convergence is expected to have slowed down due to resistance associated with thickened lithosphere": no ! it is too high ! India slows down in the early collision time (Guillot et al., 2008). There is a problem with your Indian lithosphere budget.

Line 221: in your model there is no upwelling as there is no downwelling, your detached lithosphere did not sink.

Line 236: "Implications for Tibetan evolution" your data cannot constrain such a scenario, which is not new in any case (Chung et al., 2005), your data add more problems than constraints to this model: no convective removal, as no sinking !

Reviewer #2 (Remarks to the Author)

The goal of this well-written manuscript is to interpret the author's previously published wavespeed model for the crust and mantle of eastern Asia "as they relate to the post-collision fate of the Indian, Tibetan and Asian mantle lithospheres, and to better understand the connection between lithospheric evolution and the surface expressions of the plateau uplift and volcanism" (lines 75-78). The manuscript presents little that is not in previous published journal papers but does a good job of synthesizing the published work and reconciling a number of past suggestions which formerly seemed at odds with each other.

I only have a couple of comments concerning the manuscript.

There is a sentence (lines 123-125) which is correct but might be misleading in that the earthquakes do terminate at depths shallower than 150 km as stated -- they actually terminate at a much shallower depth of about 100 km.

There are statements at a number of places in the manuscript (e.g., lines 173-174; 195-196) about the foundering of continental lithosphere. After reading the manuscript I did not have a clear idea if the authors were referring to foundering of the whole lithosphere (i.e., crust and mantle) or delamination and foundering of the mantle portion of the continental lithosphere. The buoyancy of the continental crust makes long-term subduction of the whole continental lithosphere almost impossible; where continental crust has been taken down to large depths (e.g., Dabie Mountains), its buoyancy brings it back to the surface.

The authors might find McKenzie and Priestley, *EPSL*, 435, 94--104, 2016 pertinent to their story.

Reviewers' comments:

Reviewer #1 (Remarks to the Author) Dr. Anne Replumaz:

This paper presents the results of a new tomographic model of Tibet (EARA2014), using a new method, called adjoint tomography, taking advantage of multiple seismic phases, as P and S body waves and Love and Rayleigh surface waves, recorded by a wide-aperture dense array. Compared to traditional tomographic methods which heavily rely on “crustal corrections”, adjoint tomography has the advantage of incorporating 3-D crustal structure in the initial model and iteratively updating it with both body and surface waveform information, thereby effectively minimizing crustal contamination of the mantle in the final images.

The new model seems valuable, improving the resolution of the tomographic images beneath Tibet. But I am not a specialist and I cannot judge the pertinence of the method used, or the choice of parameters in the modeling.

In the last decades, global tomography became a very powerful tool to explore the roots of the mountain ranges. Having a new tomographic model with a better resolution could be of great interest for the community. For example a new model to map precisely the underthrusting front of Indian lithosphere, interpreted in this paper approximately along the Jinsha suture (Line 85), but interpret to be along the Bangong suture from body waves tomography (e.g. Replumaz et al., 2013), is of great interest.

But unfortunately I learnt more by reading the supplementary material than the paper itself. In the paper, the figures are too small or too complex to see the data!

We appreciate the positive comments on our model, which we expect to better constrain the amplitudes and patterns of the shear wave speed anomalies in the crust and the upper mantle. Our model reveals a T-shaped high shear wave speed anomaly, which is a new feature that has not been identified by previous P-wave models.

A1. Figure 1 is too complex, many contours from tomography, petrological data, faults, sutures, a lot!

We took out the minor faults and kept the major faults and sutures for reference. The map now only has three contour lines from our tomographic model and two groups of petrological data points, which we think are essential to keep for showing the spatial correlations that support our proposed tectonic interpretation. We have also revised the figure caption accordingly.

A2. Figure 2: too small, with the green circles masking the crustal part of the section, many rifts indicated by red arrows but impossible to see if they can be seen on the tomographic sections. It is said in the text that “Seismicity is distributed along the interpreted upper interface of Indian lithosphere and terminates at depths shallower than 150 km” but it is impossible to see it on the figure. Even on figure S5 it is not clear.

We added one figure (the new Figure 2 in the main text) to show map views of shear wave speed anomalies at different depths in the upper mantle. We also split the three vertical cross sections in the original Figure 2 into two figures (the new Figures 3 and 4) to better show seismic features. In order to make the crustal part of our tomographic model clear to see, we plot the absolute shear wave speeds with seismicity superimposed in the upper 250 km with 4× of vertical exaggeration. We also modified Supplementary Figure 5 accordingly to make it clear.

A3. Similarly the “~200 km wide strongly high-V zones (more than 4% increases) alternate with ~100 km wide relatively weakly high-V zones (3%–4% increases)” are also not clearly visible on the figure, which is very frustrating.

Please see the new figure, Figure 3c, with $\delta \ln V_s$ contour lines of 3% and 4% superimposed on the V_s cross section along arc-parallel profile A. The strongly high-V zones (more than 4% increases) are about 200 km wide alternating with relatively weakly high-V zones (3%–4% increases) about 100 km wide beneath South-central Tibet, where the rift zones are located.

A4. Figure 3: 3D view with so much text on it (probably because the figure is not clear!), the vertical and horizontal sections are not clearly visible. On the contrary, in the supplementary material, all the figures are clear, at the right size, with separated figures for separated data, which make the interpretations of the authors clearer and more convincing.

Thanks for your suggestions. We have moved one supplementary figure, previously Figure S3, to Figure 2 in the main text to help support our interpretation. As mentioned above, we also split the old Figure 2 into new Figures 3 and 4, so the old Figure 3 (the 3-D views) becomes the new Figure 5. The vertical and horizontal sections in the 3-D view are meant for geographic reference. We would like to keep the text on the 3-D figure, as they are necessary and adequate.

A5. The other main point with the paper is that the authors use a scenario already published (Chung et al., 2005), but their new data add more problems than more constraints to this model: as you show no sinking of the detached lithosphere, it suggests that there is no convective instability of the Tibetan lithosphere, which does not supported the published scenario.

We do see the evidence of T-shaped foundering lithosphere, which is one of the main points of the paper (Fig. 4e in the main text and Fig. R1, a, b, and d in this rebuttal letter). Our resolution tests (Supplementary Fig. 1) also suggest that the T-shaped feature at deeper depth (250 km and deeper) might be artificially mapped about 50 km upwards (Supplementary Fig. 1, c and d) due to the vertical smearing effect of body wave resolution at this depth range. That's probably why we didn't see clear vertical separation between the foundering lithosphere and the interpreted underthrusting India lithosphere shallower than 250 km. However, based on our speculated scenario (Fig. 6), the foundering lithosphere sinks about 250 km in the past 25 Ma, which suggest a sinking rate of about 10 mm yr^{-1} .

A6. In the budget of the Indian lithosphere related to this scenario about half of India is missing. The authors should evoke the deep tomographic anomalies (van der Voo et al., 1999; Replumaz et al., 2010, 2013). In this scenario, it also implies a convergence rate for India higher than estimated, considering that the convergence rate is expected to have slowed down in the early collision time (Guillot et al., 2008).

Our paper focuses on interpreting more detailed upper mantle high-V anomalies and the relevant mantle lithospheric budget since the onset of "hard" continent-continent collision at about 45 Ma. *van der Voo et al. (1999)* emphasize on interpreting the tomographic anomalies at depths larger than 1000 km (lower mantle) that correspond to possible subduction cessation occurring at 65–55 Ma or earlier. In their paper the shallower anomalies (e.g., zone IV at depths less than 1000 km) are only qualitatively described as Tertiary lithospheric delamination during the India-Asia collision and continued north-south convergence. Therefore, we don't think that the deep tomographic anomalies in *van der Voo et al. (1999)* are very relevant to evoke in terms of budget of the Indian lithosphere.

Different from *van der Voo et al. (1999)*, with an improved P-wave model, *Replumaz et al. (2010, Terra Nova)* investigated the high-V seismic anomalies between depths of about 450 and 900 km beneath the India-Asia collision zone, which they interpret as subducted fragments of Indian continental material. They also state that about 300–700 km of Indian lithosphere subduction accommodate the India/Asia collision since 45 Ma. First of all, their region of interest (west of 84°E) is to the west of T-shaped lithosphere remnant revealed in this study, where we don't have the best data coverage. Although we observe weakly high-V between 80°E and 85°E (the western Tibetan Plateau) in the map of $\delta \ln V_s$ at 500 km depth (Fig. 2 in the main text), we decide to be cautious and not to over-interpret our results given the relatively sparse station coverage and weakly high-V pattern. Second of all, *Replumaz et al. (2010, Terra Nova)* also mentioned in their

conclusion that further east no evidence of large-scale steep subduction found by recent seismic tomographic studies, which our tomographic study also confirms. Finally, based on their P-wave model, another study by *Replumaz et al.* (2010, Tectonophysics) suggests that further east (east of $\sim 85^\circ\text{E}$) southeastwards extrusion of Burman and Andaman lithosphere and subsequent subduction account for about 800–900 km India-Eurasia convergence. In addition, *Replumaz et al.* (2013, Gondwana Research) suggest that ~ 1300 km of India-Eurasia convergence is absorbed by Asia during the indentation process and the rest ~ 900 km assumed to be absorbed by India within the Himalayan range. And more specifically, ~ 600 km of the total ~ 1300 km of Asia lithospheric consumption is absorbed by Asian continental subduction, which is based on the observed very weakly high-V feature (AS) (Fig. R1, f–j, in this rebuttal letter, after *Replumaz et al.* (2013)) in their P-wave model. And the rest (~ 700 km), according their speculation (*Replumaz et al.*, 2013), could have been accommodated by a combination of extrusion and shallow subduction/underthrusting processes.

Let's sum up the interpretation of our model in terms of the mantle lithosphere budget during the India-Eurasia collision. Regarding the further east of Tibetan Plateau (east of 85°E), our study interprets that the earlier shortening and thickening and subsequent foundering of Tibetan Lithosphere accommodates ~ 1350 km of convergence and the later underthrusting Indian continental lithosphere located in the upper 250 km absorbs about 750 km of India-Eurasia convergence. These two components of lithospheric mantle consumption add up to a total of about 2100 km in length, which accounts for $\sim 93\%$ of the estimated total 2,250 km mantle lithosphere budget.

There is no significant difference between the mantle lithospheric budget we use (total 2,250 km, 93% of which, ~ 2100 km, is accounted for from our tomographic image, ~ 1350 km from the Tibetan lithosphere shortening, thickening, and foundering, and ~ 750 km from India continental lithosphere underthrusting) and the budget that *Replumaz et al.* (2013) used (total 2200 km, only 59% of which, ~ 1300 km is accounted for from their P-wave tomographic images, ~ 700 km from India continental subduction and ~ 600 km from Asia lithosphere subduction, and the rest based on speculated hypothesis of combination of extrusion and shallow subduction/underthrusting processes).

A7. The other key point is the subduction Asian lithosphere visible on seismic profiles (e.g. Kind et al., 2002). Using global tomography to see such subduction is in fact not obvious as the asian lithosphere amplitude anomaly is weaker than the amplitude related to indian lithosphere. Nevertheless such subduction is visible in the Pwaves tomography, as a weak anomaly (Replumaz et al., 2013). You have to discuss why you do not see it in your model. You can argue that you don't see the Asian subduction because your resolution is not high enough, you cannot say that it does not exist.

We understand Dr. Replumaz's reasoning of making direct comparison between our model and Dr. Spakman's P-wave model. Although we never succeeded in requesting the digital format of the P-wave model from Dr. Wim Spakman, we manage to extract the images of the cross sections from *Replumaz et al.*'s (2013) paper. We also plotted our model side-by-side as comparison (Fig. R1 in this rebuttal letter).

We can see that Dr. Spakman's P-wave model indeed shows very weakly high-V pattern for the interpreted Asian lithosphere (*Replumaz et al., 2013*). However, the P-wave anomaly amplitude is really weak and the shape of such weakly high-V can be simply due to inversion artifact. Especially Fig. R1i (in this rebuttal letter) shows that Dr. Spakman's P-wave model at shallow depths (< 250 km) provides no evidence of low-V to high-V downward jump across the interface indicated by the green line (*Kind et al., 2002*), yet the weakly high-V anomalies at much deeper mantle (at 800 km–1000 km depths) are interpreted as Asian lithosphere in *Replumaz et al.'s* (2013) paper. On the other hand, our model (Fig. R1d in this rebuttal letter) shows the possible low-V to high-V downward jump across the interface indicated by green line (*Kind et al., 2002*), which we speculate as an internal interface within Tibetan lithosphere rather than its upper interface. We also observe very weakly high-V anomalies at the top of the lower mantle between MFT and TS where *Replumaz et al.* (2013) suggested the detached Asian lithosphere is located. However, compare to more robustly constrained T-shaped high-V with much stronger high-V anomaly amplitude (2% to 5% increases), we prefer not to interpret the weakly high-V (less than 1% increase) in our study. Therefore, we can't say the Asian subduction exists based on our study, not because that our image resolution is not high enough, but because our model does not robustly lead to a definitive conclusion.

In addition, Dr. Spakman's P wave model only use P body wave travel time, and this study uses not only body waves (P, S, depth phases, SS, and PP) but also surface waves, which gives more robust constraints of crustal and shallow mantle structure (upper 250 km), not to mention that this study accounts properly the 3-D crustal structure and being able to update the crustal model too throughout the iterative inversion. That's why we see more detailed structure in the upper 250 km, and also quite different geometry of high-V at the deeper depths.

Figure R1. Comparison between shear wave speed anomalies ($\delta \ln V_s$) from model EARA2014 (Chen *et al.*, 2015, JGR) (left panel) and P wave speed anomalies from the previous P-wave model (right panel) after Replumaz *et al.* (2013). In the left panel cross sections, white lines represent $\delta \ln V_s$ contour levels from -4% to -2% and from 2% to 4% at 1% intervals, black dashed lines mark a depth of 250 km and the 410 - and 660 -discontinuities, and black solid lines delineate the Moho from CRUST2.0. In the right panel cross sections (Replumaz *et al.*, 2013) IN is interpreted to be related to Indian continental slab, CR to Indian Craton, and AS and TI to Asian continental slabs. For reference, the annotations of MFT (Main Frontal Thrust), TS (Tsangpo Suture), BS (Bangong Suture), and ATF (Altyn Tagh Fault) in the right panel cross sections are also plotted in the cross sections through model EARA2014 (left panel). Magenta dashed lined box in each of the right panel cross sections shows the lateral and depth range of model EARA2014 plotted on the left for side-by-side comparison.

A8. To conclude, I really think that the new tomographic model is highly valuable for the community, but I found the paper as it is presented not enough focus on the new data. The authors should first clearly describe the new model, using the clear figures of the supplementary material. I suggest to compare more deeply their results with existing models, in particular P wave global tomographic models, maybe doing more synthetic tests to show the resolution of their model. And maybe less focus on a model already published, and not very well constrain.

We appreciate the positive comments on our tomographic model. We have updated our figures to make them clear based on the suggestions here. We have compared very extensively with existing models in a previous publication (their Figs. 7 and 9, their Supplementary Figs 8 and 9 in *Chen et al.*, 2015, JGR) and with the P wave model used in previous studies (Fig. R1 in this rebuttal letter and Supplementary Fig. 2). Our model is different from other models due to different data coverage and inversion technique. In this paper, we would like to focus on the features that are new and are robustly constrained in our model. The resolution test (Supplementary Fig. 1 in this manuscript) and previous model quality assessment in *Chen et al.* (2015, JGR) are sufficient for showing the robustness of our interested feature. Further synthetic model tests would require substantial amount of computing time and resources, which very unfortunately can't be afforded at this moment.

Specific comments

A9. Line 85: “*underthrusting front of Indian lithosphere*” is a result of your new model to discuss in detail, not a fact! You have to compare with P waves global tomographic models (e.g. *Replumaz et al.*, 2014)

We changed the wording in line 85 (now lines 87–90) and discussed the underthrusting front of Indian lithosphere in the discussion (please see lines 127–151). We also added Supplementary Fig. 2 to compare our model with the global P-wave model (*Replumaz et al.*, 2014).

A10. Line 86: “*Southern Tibet is divided into three sub-regions from west to east*” I don't find this description pertinent, there is no division of Tibet, there is a deep positive anomaly beneath the central part of Tibet.

We'd like to keep this division of Tibet for convenience of describing our results as well as discussion (please see lines 87–93).

A11. Line 110: *also visible on the synthetic tests (S1).*

Thanks for pointing it out. Supplementary Fig. 1 is now referenced in line 120 of the main text.

A12. Line 118: *again the northern extent of the indian lithosphere is something to discuss!*

Please see the reply to point A9 above.

A13. Line 123: “*Seismicity is distributed along the interpreted upper interface of Indian lithosphere and terminates at depths shallower than 150 km*” impossible to see something with a so small figure! even on figure S5 it is not clear...

Please see the reply to point A2 above.

A14. Line 131: *reasoning too fast, it implies that it is a crustal process, not a mantle driven process.*

We changed it accordingly, please see lines 164–171.

A15. Line 135: *“~200 km wide strongly high-V zones (more than 4% increases) alternate with ~100 km wide relatively weakly high-V zones (3%–4% increases) (Fig. 2a).” impossible to see, which is very frustrating...*

Please see the reply to point A3 above.

A16. Line 145: *“lithospheric structure of Tibet resembles that of Archaean and Proterozoic cratons, except with a hotter and thicker crust at present” i really don't think that the comparison is pertinent! it has to be removed.*

We think that it is an important observation to discuss (McKenzie and Priestley, 2008 & 2016). However, we agree that it is not a key point in our study, so we removed from the abstract the part regarding the “possible prototype of modern craton formation”, and only kept it in the discussion.

A17. Line 149: *“the T-shaped high-V structure beneath South-Central Tibet has a less obvious origin”: i agree, not obvious at all... I am not convinced by your interpretation (see discussion), you have to compare with the Pwave global tomography, showing a very different geometry, leading to a different interpretation.*

Please see Fig. R1 (in this rebuttal letter) for the comparison between the P-wave model (Replumaz et al., 2013) and our shear wave speed model. Again, The P-wave global tomography model (Replumaz et al., 2013 & 2014) has very limited vertical resolution in the upper mantle due to the fact that they only use body wave travel time without any surface wave constraint. However, we don't have resolution in the lower mantle. Please also see the replies to points A5–A7.

A18. Line 156: *“Japan and Izu-Bonin convergent margins is associated with abundant deep-focus seismicity and interpreted as subducting oceanic lithosphere, the high-V structure beneath South-Central Tibet is a much broader feature and completely lacks seismicity.” yes for sure it is not an oceanic subduction, you should also remove this not pertinent comparison.*

We would like to keep it because it supports our argument that it is not oceanic subduction.

A19. Line 159: *“T-shaped high-V structure is unlikely subducting oceanic lithosphere and the portion of Indian oceanic lithosphere probably already sank into the lower mantle.” Yes it has been shown by interpreting the deep tomographic anomalies (van der Voo et al., 1999; Replumaz et al., 2010), it should be evoked here, and more taken into*

account by the authors in their reasoning for the lithosphere budget (Replumaz et al., 2013).

Point taken, the relevant references are cited and please also see the reply to point A6 regarding the lithosphere budget.

A20. Line 171: “length estimate of about 1,354 km (supplementary text S3)” yes but you don't interpret it as indian but Tibetan lithosphere, so it is still missing half of India ! there is no length estimation in S3 but figure 4.

Please see the reply to point A6 and changes in lines 205–215.

A21. Line 181: “no obvious evidence of south dipping Asian lithosphere under Northern Tibet is shown in the arc-normal cross section (Fig. 2c and Supplementary Text S3), which is consistent with previous tomographic results.” It is really a key point here: the Asian lithosphere subduction is not obvious because the Asian lithosphere amplitude anomaly is weaker than the amplitude related to Indian lithosphere. Nevertheless Kind et al (2002) showed the subduction of the Asian lithosphere. Such subduction is visible in the Pwaves tomography, as a weak anomaly (Replumaz et al., 2013). You have to discuss why you do not see it in your model. You can argue that you don't see the Asian subduction because your resolution is not high enough, you cannot say that it does not exist.

Please see the reply to point A7.

A22. Line 186: “the Tibetan lithosphere is considered to be hotter and, as result, most likely rheologically weaker than the colder Indian lithosphere, we speculate that the Tibetan mantle lithosphere is more prone to thicken” reasoning not clear, it is hotter now as it has thicken because it was weaker.

According to Chung et al., (2009), Tibetan lithosphere was part of oceanic arc, resulted from intense basaltic underplating and subsequent remelting during the Late Cretaceous and Eocene time, related to the Neotethyan subduction processes including breakoff of the subducted slab at ca. 50 Ma in the early stage of the India–Asia collision. These processes were responsible for not only the juvenile crust formation but also for the creation of a thermally softened lithosphere in the southern Lhasa terrane.

A23. Line 193: “Due to the Rayleigh-Taylor instability, the viscous lower part of thickened Tibetan mantle lithosphere can initially drip” It is another key point in your reasoning here: how do you explain that it is dense enough to drip but not to sink in the mantle and stay just below the Indian lithosphere for 30 Ma?

It was sinking but at really slow speed ($\sim 10 \text{ mm yr}^{-1}$) and the other possibility is that the 660-km discontinuity acts as a barrier to prevent the T-shaped structure from sinking to the lower mantle due to its large viscosity jump across.

A24. Line 199: there is also potassic magmatism between 40 and 30 Ma, much wider than the ultra-potassic you show, you cannot choose between magmatism pulse like that.

The potassic magmatism in the Qiangtang terrane between 40 and 30 Ma was explained in *Chung et al. (2005)* to be associated with slab rollback, back-arc extension, and basin formation. We are more interested in ultrapotassic especially adakitic magmatism, as it is an indication of lithospheric thickening, subsequent lithospheric foundering and upward counterflow of the hotter asthenosphere (*Chung et al., 2009*).

A25. Line 204: maybe, but the plateau is much wider than the deep anomaly, so it cannot explain the formation of the plateau.

Our model is more robustly constrained beneath Central and Eastern Tibet. Future new models with more data coverage of Western Tibet will better explain the formation of the entire plateau.

A26. Line 206: "The continued sinking of foundering lithosphere in the upper mantle": it is one of the main problem of your model: the sinking is very limited. How could you have a convective instability which reheated the crust, without sinking down to the lower mantle?

Please see the reply to point A23.

A27. Line 218: "It is slightly higher than the current ongoing convergence rate of ~20 mm/yr between India and Indus-Yarlung Suture but remains a reasonable estimate as the convergence is expected to have slowed down due to resistance associated with thickened lithosphere": no! it is too high! India slows down in the early collision time (Guillot et al., 2008). There is a problem with your indian lithosphere budget.

Please see the replies to point A6.

A28. Line 221: in you model there is no upwelling as there is no downwelling, your detached lithosphere did not sink.

Please see the replies to point A5 and point A23.

A29. Line 236: "Implications for Tibetan evolution" your data cannot constrain such a scenario, which is not new in any case (chung et al., 2005), your data add more pb than constrain to this model: no convective removal, as no sinking!

Please see the reply to point A5.

Reviewer #2 (Remarks to the Author):

The goal of this well-written manuscript is to interpret the author's previously published wavespeed model for the crust and mantle of eastern Asia "as they relate to the post-

collision fate of the Indian, Tibetan and Asian mantle lithospheres, and to better understand the connection between lithospheric evolution and the surface expressions of the plateau uplift and volcanism" (lines 75-78). The manuscript presents little that is not in previous published journal papers but does a good job of synthesizing the published work and reconciling a number of past suggestions which formerly seemed at odds with each other. I only have a couple of comments concerning the manuscript.

B1. There is a sentence (lines 123-125) which is correct but might be misleading in that the earthquakes do terminate at depths shallower than 150 km as stated -- they actually terminate at a much shallower depth of about 100 km.

The vast majority of the earthquakes indeed terminate at a much shallower depth of about 100 km (Priestley *et al.*, 2008, GJI). However, along profile B (Fig. 4c) we do see one earthquake located at a depth of about 150 km that is also close to our interpreted upper interface of underplating Indian lithosphere. This earthquake's location information is from EHB event catalogue.

EVENTID: 12477266

AUTHOR: EHB

DATE: 1980-08-11

TIME: 07:33:29.21

LAT: 35.6200°

LON: 77.5540°

DEPTH: 140.0 km

It is possible that this earthquake belongs to southward subducting Pamir slab, annotated as AL (Asian Lithosphere) in Fig. 4c, because there is a large gap between this deep event and much shallower crustal seismicity along the interface of IL (Indian lithosphere), but it is closer to the intermediate-depth seismicity along the interface of AL. We made some changes according in the text, please see lines 152–158.

B2. There are statements at a number of places in the manuscript (e.g., lines 173-174; 195-196) about the foundering of continental lithosphere. After reading the manuscript I did not have a clear idea if the authors were referring to foundering of the whole lithosphere (i.e., crust and mantle) or delamination and foundering of the mantle portion of the continental lithosphere. The buoyancy of the continental crust makes long-term subduction of the whole continental lithosphere almost impossible; where continental crust has been taken down to large depths (e.g., Dabie Mountains), its buoyancy brings it back to the surface.

This is an excellent point and thanks a lot for bring this up. We are referring to foundering of the majority of the mantle portion of the thickened continental lithosphere at the bottom, but with some minor part of the mantle lithosphere at the top still left behind attached to the crust above. We've added lines 208–211 to clarify what foundering means in this manuscript.

B3. The authors might find McKenzie and Priestley, EPSL, 435, 94--104, 2016 pertinent to their story.

We added this reference at a couple of places where it is pertinent. Please see lines 171 and 186.

Reviewers' Comments:

Reviewer #1:

Remarks to the Author:

The authors deeply re-write and re-do the figures according to my comments. It strengthens the paper a lot, the data are presented more clearly now, and the discussion is more complete. I recommend to accept this paper, showing new tomographic images beneath Tibet with high resolution.

Anne Replumaz